# Combining viral genetic and animal mobility network data to unravel peste des petits ruminants transmission dynamics in West Africa

Arnaud Bataille[1,2]*, Habib Salami[1,2,3]ꙩ, Ismaila Seck[4,5]ꙩ, Modou Moustapha Lo[3], Aminata Ba[3], Mariame Diop[3], Baba Sall[4], Coumba Faye[4], Mbargou Lo[4‡], Lanceï Kaba[6], Youssouf Sidime[6], Mohamed Keyra[6‡], Alpha Oumar Sily Diallo[6], Mamadou Niang[7], Cheick Abou Kounta Sidibe[7], Amadou Sery[7], Martin Dakouo[7], Ahmed Bezeid El Mamy[8], Ahmed Salem El Arbi[8], Yahya Barry[8], Ekaterina Isselmou[8], Habiboullah Habiboullah[8], Abdellahi Salem Lella[8‡], Baba Doumbia[8], Mohamed Baba Gueya[8], Caroline Coste[1,2†], Cécile Squarzoni Diaw[1,9], Olivier Kwiatek[1,2], Geneviève Libeau[1,2], Andrea Apolloni[1,3,10]

1 ASTRE, Univ Montpellier, CIRAD, INRAE, Montpellier, France, 2 CIRAD, UMR ASTRE, Montpellier, France, 3 Institut Sénégalais de Recherches Agricoles, Laboratoire National d'Elevahge et de Recherches Vétérinaires (LNERV), Dakar-Hann, Sénégal, 4 Direction des Services Vétérinaires, Dakar, Senegal, 5 FAO, ECTAD Regional Office for Africa, Accra, Ghana, 6 Institut Supérieur des Sciences et de Médecine Vétérinaire, Dalaba, Guinea, 7 Laboratoire Central Vétérinaire (LCV), Bamako, Mali, 8 Office National de Recherches et de Développement de l'Elevage (ONARDEL), Nouakchott, Mauritania, 9 CIRAD, UMR ASTRE, Ste-Clotilde, La Réunion, France, 10 CIRAD, UMR ASTRE, Dakar Hann, Sénégal

ꙩ These authors contributed equally to this work.
† Deceased.
‡ Unavailable.
* arnaud.bataille@cirad.fr

**Data Availability Statement:** Genetic data have been deposited in GenBank (accession numbers MT072445-MT072545). Other relevant data are

## Abstract

Peste des petits ruminants (PPR) is a deadly viral disease that mainly affects small domestic ruminants. This disease threaten global food security and rural economy but its control is complicated notably because of extensive, poorly monitored animal movements in infected regions. Here we combined the largest PPR virus genetic and animal mobility network data ever collected in a single region to improve our understanding of PPR endemic transmission dynamics in West African countries. Phylogenetic analyses identified the presence of multiple PPRV genetic clades that may be considered as part of different transmission networks evolving in parallel in West Africa. A strong correlation was found between virus genetic distance and network-related distances. Viruses sampled within the same mobility communities are significantly more likely to belong to the same genetic clade. These results provide evidence for the importance of animal mobility in PPR transmission in the region. Some nodes of the network were associated with PPRV sequences belonging to different clades, representing potential "hotspots" for PPR circulation. Our results suggest that combining genetic and mobility network data could help identifying sites that are key for virus entrance and spread in specific areas. Such information could enhance our capacity to develop locally adapted control and surveillance strategies, using among other risk factors, information on animal mobility.

within the manuscript and its Supporting Information files.

**Funding:** Ar. B., O.K., H.S. and G.L. were supported by a grant from the European Commission Animal Health and Welfare European Research Area Network (https://ec.europa.eu/food/animals/health_en) for the IUEPPR Project "Improved Understanding of Epidemiology of PPR", in the framework of ANIHWA 2013. Ar. B., O.K. and G.L. were supported by a grant (SI2.756606) from the European Commission Directorate General for Health and Food Safety awarded to the European Union Reference Laboratory for Peste des Petits Ruminants (EURL-PPR). Ar.B., A.A., M.M.L. were supported by the European Commission through the International Fund for Agricultural Development (grant number 2000002577) and the CGIAR research program on Livestock. We thank all donors who support the work of the CGIAR research program on Livestock through their contributions to the CGIAR trust fund. The funders had no role in study design, data collection and analysis, decision to publish, or preparation of the manuscript.

**Competing interests:** The authors have declared that no competing interests exist. Authors Mbargou Lo, Mohamed Keyra, Abdellahi Salem Lella were unable to confirm their authorship contributions. On their behalf, the corresponding author has reported their contributions to the best of their knowledge.

## Author summary

As animals move so do viruses. The viral disease peste des petits ruminants (PPR) has a major impact on the livelihood of sheep and goat farmers across Africa, Middle-East and Asia. A global PPR eradication campaign is underway, but extensive movements of infected animals impede control efforts in many regions, such as West Africa. Here we show for the first time that PPR virus genetic data can be combined with information on animal mobility to identify routes of PPR circulation in Senegal and neighbouring countries. Such information can be used to design more efficient disease surveillance and control strategies adapted to local livestock farming practices.

## Introduction

Peste des petits ruminants (PPR) is a highly pathogenic disease that mainly affects small domestic ruminants (sheep and goats) across Africa, Asia, and the Middle East [1,2]. PPR spreads rapidly among susceptible animals mostly through direct contact, with mortality rates sometimes reaching 90% in infected flocks [3]. PPR represents a threat to food security and to the livelihoods of smallholder farmers, with economic losses estimated at between US$1.5 and 2.1 billion per year [4]. PPR is now the target of a global eradication campaign led by the World Animal Health Organisation (OIE) and the Food and Agriculture Organisation (FAO) [5]. Despite the availability of efficient and cheap vaccines against PPR, control of this disease is complicated, notably because of extensive movement of animals through within- and trans-boundary trade, and seasonal transhumance, which is often poorly monitored in regions where PPR is endemic [6,7]. For example, importing infected sheep from abroad for the purpose of fattening was at the origin of the re-emergence of PPR in Morocco in 2015 just a few years after the disease had been completely eradicated [8].

Mobility plays an important role in the West Africa [9]. In arid and hyper arid areas, animals are constantly on the move looking for water sources and better grazing areas. Due to climatic harsh conditions every year, several thousand animals move from the arid areas of Mauritania, Mali, Burkina Faso and Senegal, towards the humid and greener areas of the coastal countries (Great Transhumance). In parallel to these long international transhumance movements, animals move away from agricultural production zones to other regions in the country so that the land can be used for cultivation (Short Transhumance) [10]. Furthermore, due to the absence of stocking facilities and slaughterhouses, most animals are sold alive and the majority are moved on foot from one location to another [11]. As the animals move so do viruses. As observed with other diseases, long range transhumance can re/introduce a pathogen in naïve areas, while commercial movements could harmonise the epidemiological situation in the country [12,13].

Better information on livestock mobility, as recommended by OIE, could help develop PPR surveillance and control strategies that are better adapted to regional and national risks of disease spread through livestock movement [11,14]. Such data can be collected using low-budget and low-technology surveys which are easy to put in place in regions where PPR is endemic [11]. However, to our knowledge, to date, there is limited direct evidence of the importance of specific animal routes or livestock markets in PPR spread. Insights into PPR transmission dynamics have often come from phylogenetic studies based on short genetic portions of its causative agent, Peste des petits ruminants virus (PPRV; e.g. [6,8,15]). PPRV is an RNA virus of the genus *Morbillivirus* (species name *Small ruminant morbillivirus*) that can be separated

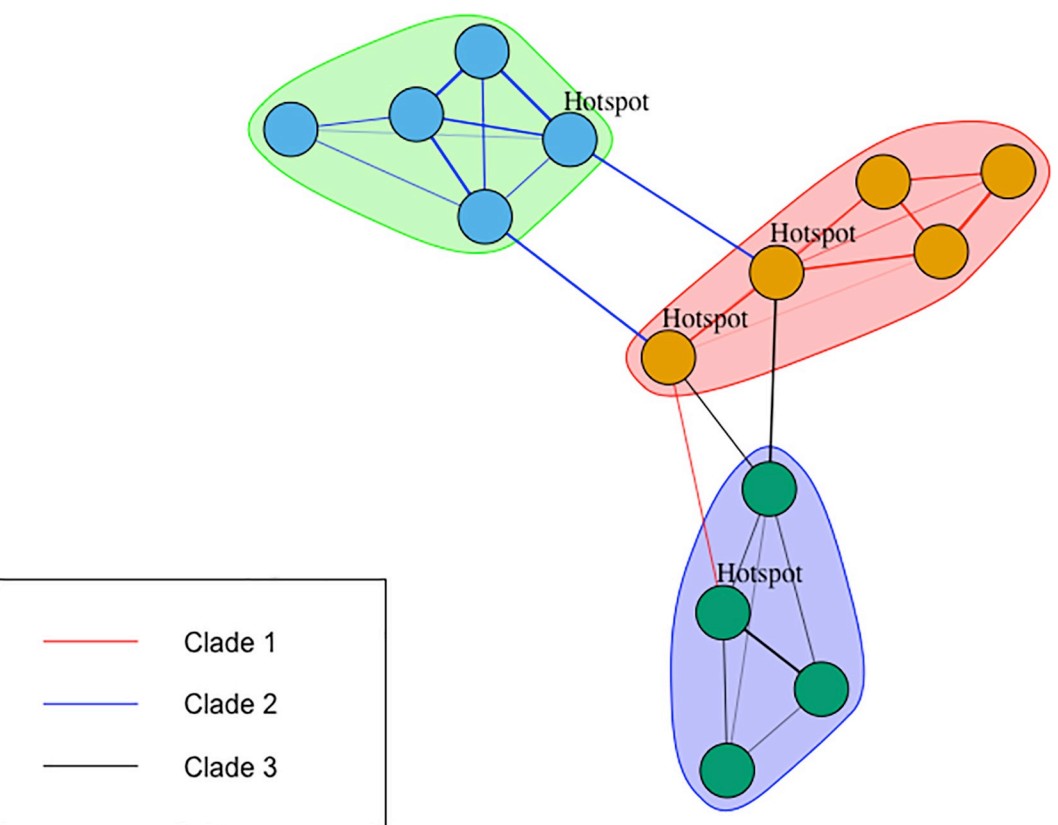

**Fig 1. Graphical representation of the relation between network communities and virus transmission.** Colored shades group sets of nodes belonging to the same community in the network. Colored links represent different virus strains in circulation.

into four distinct genetic lineages (lineages I-IV). All the lineages are present in Africa whereas in the Middle East and Asia, animals are mostly infected by Lineage IV PPRV strains [16]. Phylogenetic analyses can be powerful tools to unravel the transmission dynamics of pathogens, especially for fast-evolving RNA viruses [17]. However complex spatially-informed or time-scaled phylodynamic tools have only been developed to study infectious disease epidemics and are not really suitable for endemic situations where circulating strains have multiple origins and unbiased sampling is difficult to put in place [18].

Animal mobility can be interpreted as networks, with underlying dynamics that favours the interactions among specific set of nodes thus creating structures, i.e. communities (Fig 1). In the case of animal trade, communities can be formed because locations belong to same commercial chain, or are situated along the same axis of movement. In the example represented in Fig 1, coloured shades around sets of nodes indicate communities in the network, and coloured links indicates that specific strains are carried by animals moving between locations. Because members in the same communities tend to interact mostly among themselves, the diffusion of one virus strain could be facilitated once introduced in the community. Depending on their position in the network, certain nodes could be exposed to multiple strains ("hotspot"). The identification of these hotspots could be relevant in identifying possible sentinel nodes in the network.

Here, we explore the strength of the relation between PPR virus genetic data from PPR virus and livestock mobility in Senegal and neighbouring countries. Both Senegal and Mauritania maintain a system of animal mobility certificates. These certificates provide a wealth of

data quite unique in Africa [14] and we used them for the network analyses in this study. Data on PPR virus circulation, including genetic data, is limited in most African countries, notably due to budget constraints, confounding symptoms with other diseases (Bluetongue, ORF disease, Contagious caprine pleuropneumonia etc.), and the limited reporting of active disease by communities endemically burdened by multiple small ruminant diseases [19,20]. For the present study, potential PPR infections were investigated by Senegal National Veterinary Services between 2010 and 2014 to gather one of the most complete PPRV genetic datasets ever obtained from one country, which was complemented with genetic data from neighbouring countries. We used phylogenetic analyses to identify clades of PPRV sequences of common origin. We searched for correlations between the genetic distances that separate PPRV strains of the same clade and animal mobility network distances. We also assessed whether variations in PPRV genetic diversity among sites sampled could be linked with characteristics of the animal mobility network. The implications of our results for PPR surveillance and control strategies are discussed.

## Results

### PPR virus genetic diversity and clustering in the study area

PPR virus sampling in Senegal was carried out from 2010 to 2013 across departments based on local veterinary reports of small ruminants showing clinical signs suggestive of PPR infection, with a total of 42 sites surveyed (S1 Table). In addition, samples were obtained from veterinary services in Guinea, Mali and Mauritania. A total of 865 samples were collected between 2010 and 2014, mainly in Senegal (825), but also in Guinea (3), Mali (31), Mauritania (7). Of those, 95 samples (Senegal = 80, Guinea = 3, Mali = 9, Mauritania = 3) were confirmed as PPR infection by RT-PCR (S1 Table). A partial nucleoprotein (N) gene sequence was obtained from most Senegal samples (74) and from all the samples from the three other countries (S2 Table). The phylogenetic analysis based on partial N gene sequences showed that all PPRV samples obtained belonged to the PPRV genetic lineage II, except for one sample from Sosorona in Mali, which clustered with lineage I (S1 Fig and S1 Data). The complete nucleoprotein (N) and hemagglutinin (H) genes were then sequenced in a selection of samples to allow for more refined phylogenetic analyses. Complete nucleoprotein (N) and hemagglutinin (H) gene sequences were obtained from 37 PPRV samples from Senegal, all lineage II samples from Mali, and all samples from Guinea and Mauritania (S2 Table and S2–S4 Data). Bayesian inference and Maximum Likelihood phylogenetic analyses based on concatenated N and H gene sequences showed that most sequences obtained during this study clustered in a discrete number of well-defined clades (posterior probability/bootstrap values > 80%; Figs 2 and S2). Six clades grouped between three and ten sequences from different regions and were of particular interest for this study (Figs 2 and 3). Four of these clades included PPRV sequences obtained in different countries. Clade 2 grouped three samples from Mauritania and the Matam region in Senegal. All PPRV sequences from Guinea clustered in clade 3 with six samples from Senegal. Clades 5 and 6 contained three to four sequences from Senegal and Mali (Figs 2 and 3). Pairwise genetic distance between samples belonging to the same clade (mean distance = 0.30%) was significantly lower than the distance between samples placed in different clades (mean distance = 0.52%; Wilcoxon test, W = 10823, $p < 0.0001$). Clades grouped samples from the same collection year more often than expected by chance (Fisher's exact test; p-value < 0.001).

### Animal mobility network in the study area

We used data on animal mobility collected through two ad-hoc activities in Senegal and Mauritania and described in previous studies [11,14,21]. For both countries, we considered data

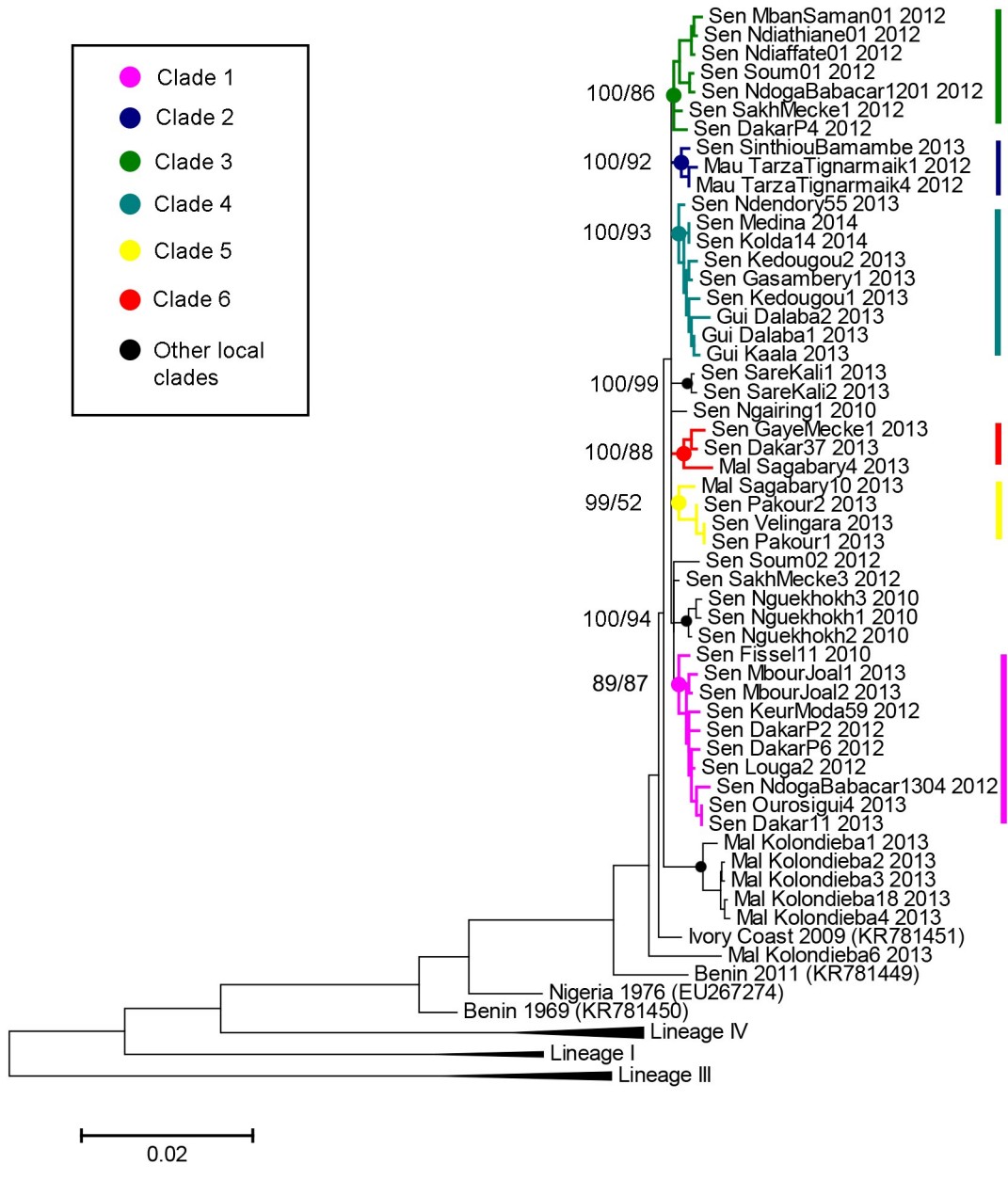

**Fig 2. Phylogenetic analysis based on concatenated PPRV N and H complete coding regions.** Phylogenetic tree constructed using a maximum likelihood inference method showing the relationship based on N and H gene sequences of peste des petits ruminants virus (PPRV) samples collected in Guinea, Mali, Mauritania and Senegal (see S1 Table for details). Genetic clusters of interest to this study are indicated by coloured branches. The numbers at the nodes indicate support from posterior probability/bootstrap values (> 50%) obtained with Bayesian inference and maximum likelihood methods, respectively.

collected in 2014 on small ruminant movements. We used a complex network approach where villages corresponded to nodes and movements between nodes were represented by links weighted either by the number of animals exchanged (volume) or by the number of exchanges between the nodes (frequency). The network obtained consisted of 270 nodes and 507 links. Nodes were mostly concentrated in Senegal, but their geographical extent ranged from Guinea-Bissau and Guinea (in the south) to Mauritania (in the north) and Mali (in the east).

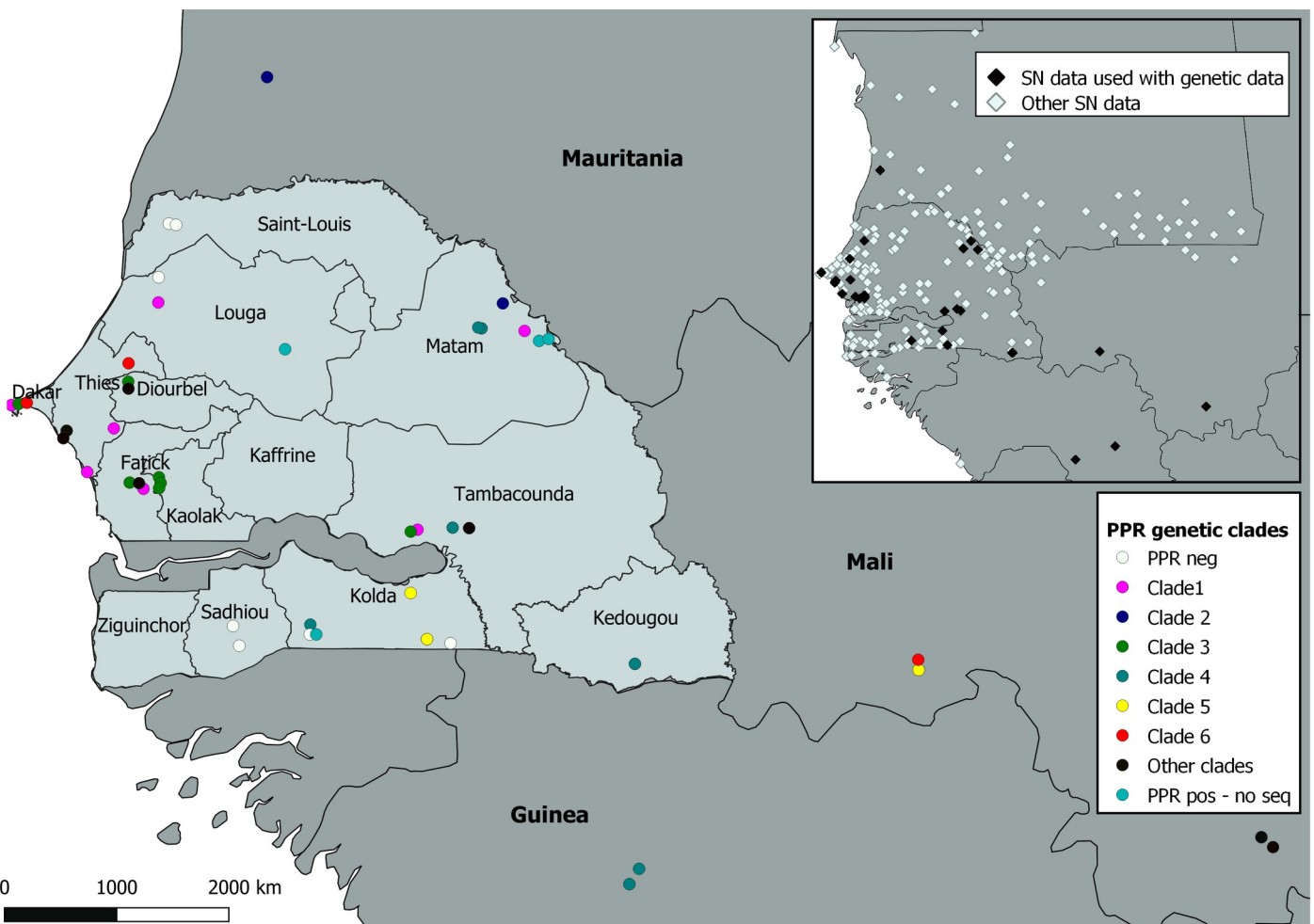

**Fig 3. Geographical locations of collected peste des petits ruminants virus (PPRV) samples (main figure) and network nodes distribution (inset).** The colours correspond to genetic clades identified in this study. Labels in the main figure correspond to names of regions in Senegal and neighbouring countries. Labels in the inset indicate network (SN) data used in combination with genetic data in the study. The base layer for the map used in the figure was obtained at http://www.gadm.org.

On average, movements between locations were not frequent, i.e., they occurred only four times per year with 6,852 animals transported. However, the network showed a high degree of heterogeneity in both frequency and volume. The network contained five weakly-connected components, including one very large one, formed by 262 nodes, and four made of only pairs of nodes. Both the diameter and the mean distance of the network, measured on the largest connected component, were low (respectively 9 and 3.8), indicating the presence of hubs that could facilitate the diffusion of the disease. Both the indegree ($k_{in}$) and outdegree ($k_{out}$) distributions ($P(k_{in})$, $P(k_{out})$, respectively) followed a power law

$$P(k_{in}) \sim k_{in}^{-\alpha_{in}}, \ \alpha_{in} = 1.188 \ (p \ value < 0.01), \ R^2 = 0.808;$$

$$P(k_{out}) \sim k_{out}^{-\alpha_{out}}, \ \alpha_{out} = 1.309 \ (p \ value < 0.01), \ R^2 = 0.842$$

indicating the presence of a few highly connected nodes (hubs) and many poorly connected nodes (S3 Fig). The network also exhibited a high degree of heterogeneity for the (annual)

**Table 1. Summary statistics of the association between the PPRV genetic clade and animal network communities defined using two different algorithms and different link weights.**

|  | InfoMap | Edge_betweenness |
|---|---|---|
| **No weight (basic)** |  |  |
| Number of communities | 28 | 11 |
| Corr. community size–number of strains | 0.59** | 0.70** |
| Likelihood same community–same clade | 3.08 [1.74;5.32]*** | 2.03 [1.28;3.16]** |
| Corr. community-clade | 0.76 *** | 0.59*** |
| **Movement Frequency** |  |  |
| Number of communities | 31 | 8 |
| Corr. community size–number of strains | -0.05 | 0.93*** |
| Likelihood same community–same clade | 2.51 [1.41;4.35]** | 2.03 [1.28;3.16]*** |
| Corr. community-clade | 0.70*** | 0.52** |
| **Cumulative Volume** |  |  |
| Number of communities | 31 | 16 |
| Corr. community size–number of strains | 0.57* | 0.54* |
| Likelihood same community–same clade | 1.83 [1.11;2.96] * | 2.98 [1.59;5.45]*** |
| Corr. community-clade | 0.64 ** | 0.66** |
| **Brockmann** |  |  |
| Number of communities | 29 | 10 |
| Corr. community size–number of strains | 0.46* | 0.81*** |
| Likelihood same community–same clade | 2.95 [1.68;5.09]*** | 1.57 [0.98;2.48]* |
| Corr. community-clade | 0.71*** | 0.46* |

InfoMap/Edge_betweeness, type of algorithm used to calculate communities; Links between nodes of the network were weighted by the number of animal exchanged using either no weight, frequency of movement, cumulative volume, or the Brockmann distance (see main text); Number of communities detected; Corr. community size-number of strains, Pearson's correlation coefficient among community sizes and number of strains of different clades in community; Likelihood same community-same clade, Odds Ratio test for strains to be in the same community and same genetic clade with 95% confidence interval between brackets; Corr. community-clade, Fisher's exact test for correlation among communities and clades

* $p < 0.05$

** $p < 0.01$

*** $p < 0.001$.

weight distribution with almost 50% of links used to transport at most 102 animals, while less than 10% concentrated most transport, corresponding to at least 9 800 animals annually. Half the links were active just once in 2014, while only 10% were active more than 10 times over the year and mostly concentrated on the Adel Bagrou-Nara axes (S3 Fig).

We investigated if the network contained communities (i.e. subsets of nodes with denser connections than the rest of the network [22]).The presence of this type of structure indicates the preference of interactions among locations that could explain the spatial distribution of virus strains. Different community structures were obtained depending on the algorithm used, the Infomap algorithm [23], or the Edge_Betweenness algorithm [24], and the type of weight considered for the links (Table 1 and S5 Fig). The Infomap algorithm mainly detected small communities, i.e. with less than 15 nodes (S5 Fig). On the other hand, with the Edge_Between-ness algorithm, community sizes were homogeneously distributed between two and 80, except when the annual volume was used as weighting (S5 Fig). In this case, the size distribution revealed many communities made of 20 nodes. For both the algorithms used, the similarity between community partition was high as shown by the high value of the Rand Index

(Infomap = 0.92–0.96; Edge_Betweenness = 0.87–0.91). Independently of the algorithm and the weights used, the modularity for the community partition was high (above 0.3) indicating a good communities repartition. The community detection algorithms detected several communities, among which only a few contained nodes where the virus was sampled. We use the term *PPRV-community* to indicate these communities containing at least one node where PPRV was sampled, and subsequent analyses concentrated on those communities.

### Correlation analyses of genetic and animal mobility data

We tested for correlations between PPRV genetic distances and multiple other distances, some geo-related (euclidean distance, least cost distance, road distance, resistance distance) as in [25], and other network-related (netdist = number of links between two nodes, Brockmann distance = a log-distance measure using the annual average volume [26], conductance based on volume, and conductance based on frequency of movements; S5 Data). Only genetic distances between pairs of sequences within well-defined phylogenetic clades (total = 118 pairs) were used for these analyses, as they represented groups of viruses originating from a common chain of transmission events. Comparing the results provided some information on the importance of human activity (animal mobility) in comparison to environmental factors. Correlations between genetic and spatial or network-related distances taken singularly were significant (Mantel test, $r = 0.53$–$0.79$; $p < 0.001$; S3 Table). We performed partial Mantel tests to explore for potential interactions between network-related and spatial related distances [27]. Correlations between genetic and network related distances remained high after controlling for interaction with spatial distances (Mantel test, $r = 0.34$–$0.59$; $p < 0.001$; S4 Table). We separated the data in different classes according to euclidean distance, from short distance (0–78 km) to large distance (778–858 km) separating pairs of samples, to produce Mantel correlograms [28]. The analyses showed that correlation coefficients changed abruptly from highly positive ($r = 0.76$; $p < 0.001$) for short distances to the lowest negative value ($r = -0.45$; $p < 0.001$) and then remains negative for larger distances ($r = -0.36$ – $-0.06$; $p = 0.1$–$0.001$; Fig 4) hinting to the presence of an exponential decrease due to the presence of patches. Correlation coefficient values oscillated around zero when controlling for network-related distance Netdist ($r = -0.05$–$0.09$ $p = 0.1$–$0.001$; Fig 4). Similar patterns were observed with other geographical and network-related measures (S4 Fig). As these results indicated spatial structuring in the data, we performed Multiple Regression on distance matrices (MRM) analyses [29] to assess the relative importance of each explanatory distances (geographical *vs* network-related), separating the data in two subsets: short distances (containing the first 2 distances classes of the correlogram analyses, i.e. for distances $\leq 158$ km) and long distances (all other classes, i.e. distances >158 km). The results showed that, in the majority of the comparisons, geographical distances had a higher effect than network-related distances on genetic distances for sites separated by small distances ($c_{geo} = -0.0027$ – $-0.0019$, $p < 0.001$; $c_{net} = -0.0015$–$0.0006$, $p < 0.05$), but network-related distances was most important at larger distances ($c_{geo} = -0.0006$–$0.0018$, $p < 0.001$; $c_{net} = 0.0003$–$0.0032$, $p < 0.0001$; S5 Table).

We tested the hypothesis that the presence of communities could lead to some level of homogeneity in the virus strains circulating among members of the same communities, *i.e.* locations in the same community could be infected by the same or closely related PPRV strains of the same clade. For all the community partition, the modularity is in the range 0.47–0.6, well above 0.3, thus indicating a good "partition" in communities. PPRV-communities consisted of nodes located a few hundred kilometres apart (< 500 km). Clark Evans tests [30] showed that almost all communities were spatially clustered (Clark Evans test, $R<1$, $p < 0.001$). However, the largest community detected by the Edge_Betweenness algorithm,

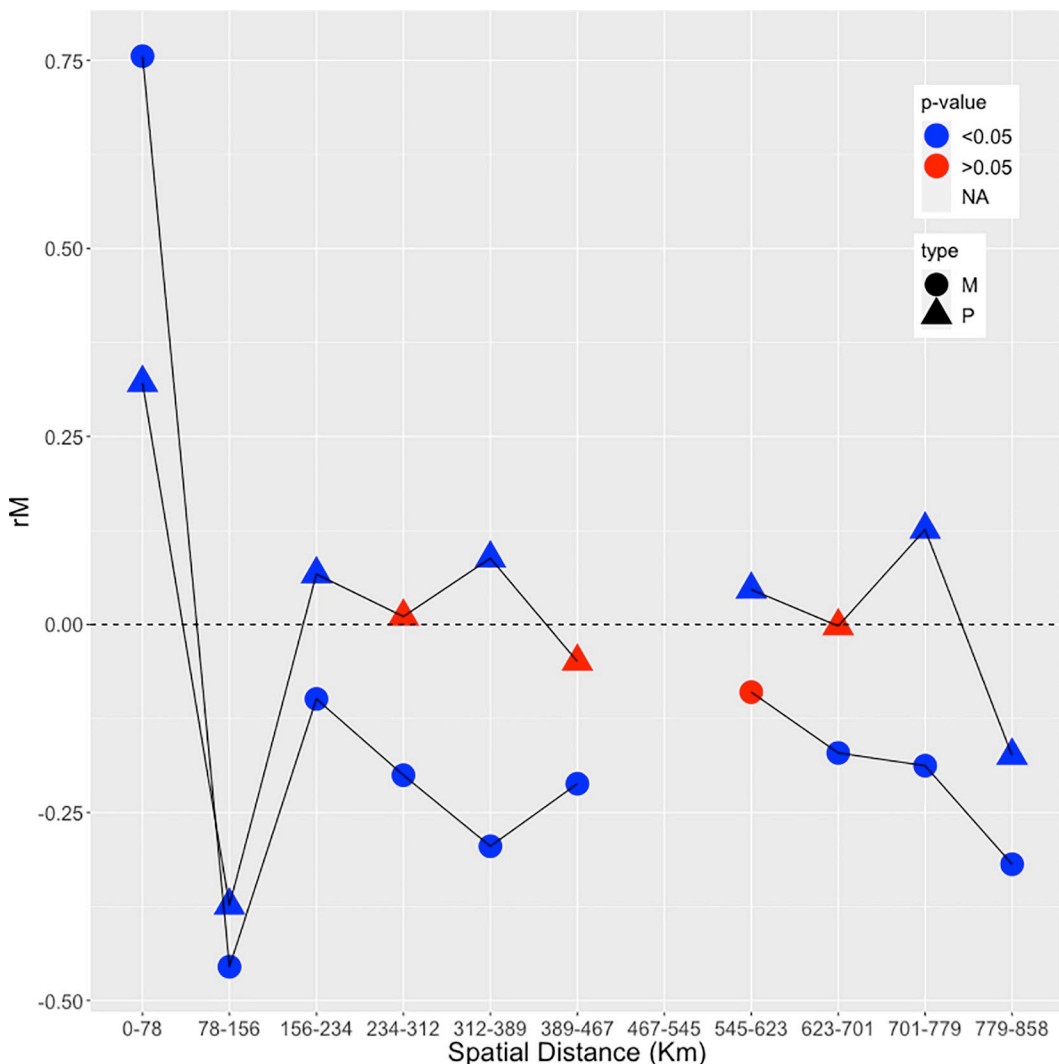

**Fig 4. Mantel Correlogram for genetic and geographic (euclidean) distance.** The value of Mantel correlation coefficient is shown for each distance class. Circles (M) correspond to result of Mantel correlogram without controlling for interaction with network distance. Triangles (P) indicate correlation analyses taking into account potential interaction with the network distance Netdist (see main text). Colors indicate significance of the Mantel tests. Results of other correlograms based on different geographical and network-related distances are shown in S4 Fig.

was not spatially clustered regardless of weighting (S6 Fig). Communities were plotted on the map of Senegal. We built a Voronoi tessellation around network nodes, and coloured each polygon according to the PPRV-community index (Figs 5 and 6). The area covered by these communities depended on the algorithm and measures used. Independently of the algorithm and criteria used for detecting communities, results of join count analyses were always positive and significant, indicating that nodes in the same communities tended to share borders and form patterns. Some patterns could be observed across the study area, notably for the south of Senegal, in the Ferlo region and in the area around Dakar in the case of the InfoMap algorithm with volume and frequency weight criteria (Fig 5). With the Edge_Betweenness algorithm, communities appeared to be less fragmented spatially with the Brockmann and frequency weight criteria (Fig 6).

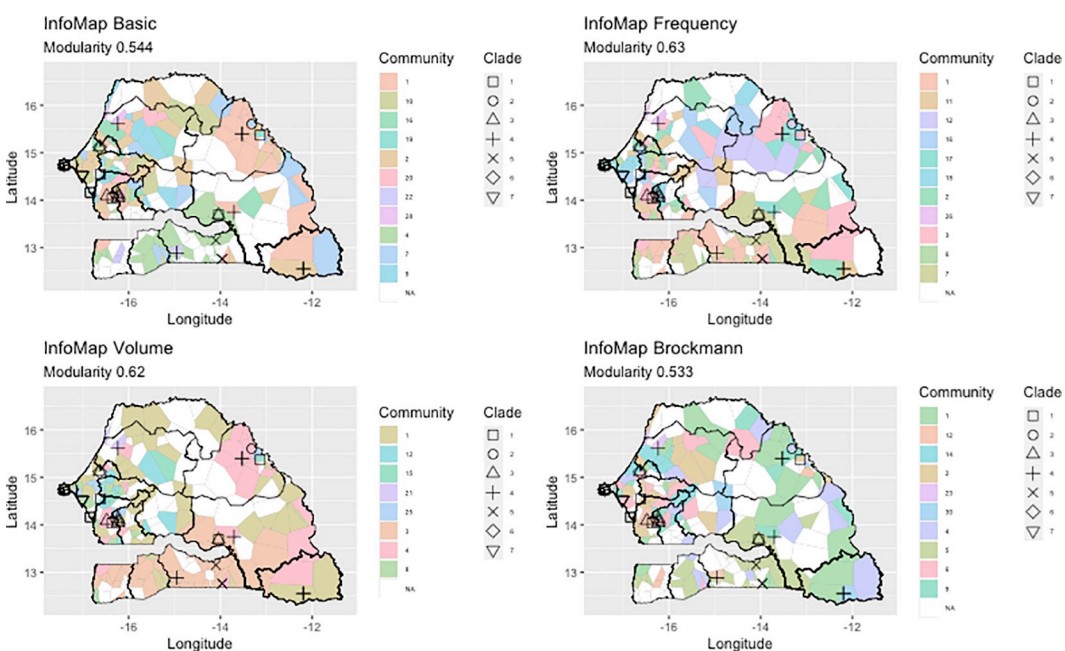

**Fig 5. Tessellation of Senegal based on the position of mobility network nodes using the InfoMap algorithm.** Each polygon, centred around nodes of the network, is coloured based on the community it has been associated with (also labelled with an id number in the legend). Only communities where at least one PPRV strain has been detected are shown in colours. Icons indicate different PPRV genetic clades based on results of the phylogenetic analyses. This figure visualizes if spatial structure is present in our network, using the community algorithm InfoMap, with different weights for the links: Basic, no weight; Frequency, the number of times the link was active; Volume, cumulative volume of animals exchanged in a year; Brockmann, using the effective Brockmann distance (see main text). The base layer for the map used in the figure was obtained at http://www.gadm.org.

In most cases, there was a strong and significant correlation between the size of the community and the number of clades in the community (Spearman correlation test, r = 0.46–0.93; $p \leq 0.05$), except when using the InfoMap algorithm weighted with the frequency of movements (r = -0.05; p > 0.05; Table 1 and S7 Fig). The Odds Ratios for two sequences to be in the same clade if they were in the same community were always higher than 1 (OR = 1.83–3.08; p < 0.001; Table 1), except for the communities found using the Edge_Betwenness algorithm and the Brockmann distance (OR = 1.57, 95% C.I. = 0.98–2.48; Table 1). The correlation between communities and clade distribution was highly significant in all cases (Fisher's exact test, Cramer V = 0.522–0.765; $p \leq 0.002$; Table 1), except for the Edge_Betweenness algorithm weighted by the Brockmann scale (Cramer V = 0.461; p = 0.03).

## Association between virus genetic hotspots and animal mobility

We identified four villages in Senegal linked with PPRV sequences belonging to more than one clade ("hotspots", Fig 7): Mbam (Fatick Region), Meouane (Thies region), Pikine (Dakar Region) and Thiara (Tambacounda Region). Mbam and Thiara are located close to the border with Gambia, and despite the fact they are not big towns, they are important centers for residents in the nearby villages and a passage point for those going to Gambia. Pikine is at the entrance of Dakar and the disembarkment point for livestock convoys before the animals can be sold in Dakar.

The presence of at least one livestock market around sampling sites was not a good predictor of virus genetic hotspots (risk ratio < 1; p > 0.05), since markets are present in the proximity of all nodes. Considering two group of nodes ("hotspot" and "monoclade", i.e. only one

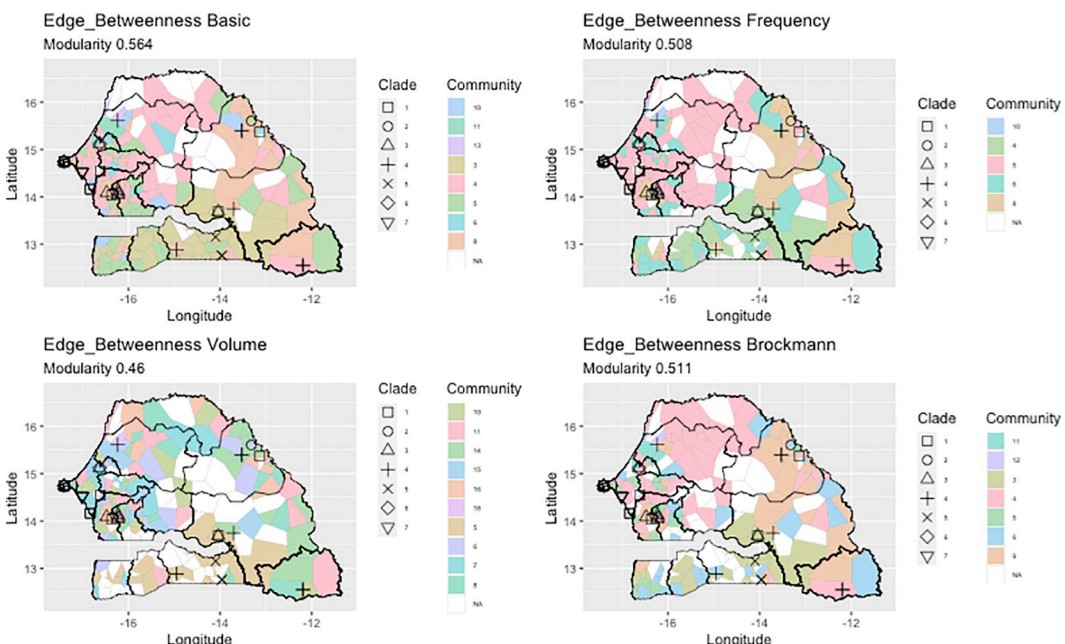

**Fig 6. Tessellation of Senegal based on the position of mobility network nodes using the Edge_Betweenness algorithm.**
Each polygon, centred around nodes of the network, is coloured based on the community it has been associated with (also labelled with an id number in the legend). Only communities where at least one PPRV strain has been detected are shown in colours. Icons indicate different PPRV genetic clades based on results of the phylogenetic analyses. This figure visualizes if spatial structure is present in our network, using the community algorithm Edge Betweenness, with different weights for the links: Basic, no weight; Frequency, the number of times the link was active; Volume, cumulative volume of animals exchanged in a year; Brockmann, using the effective Brockmann distance (see main text). The base layer for the map used in the figure was obtained at http://www.gadm.org.

genetic clade found in the village concerned), we noticed that hotspots were characterised by having, on average, more in-connections, large in-volume, higher betweenness and centrality than monoclade ones (S7 Table). Moreover, the hotspots were not the origin of animal movements (outdegree = 0 for all the hotspots), and they appeared less often as a destination of movement than monoclades (S7 Table). In terms of homophily (i.e. the fraction of links exchanged with other nodes of the same community), hotspots had a higher tendency to create links and to exchange more links with members of the same community than monoclade nodes (S7 Table).

Discriminant analysis showed that the homophily was the variable with the highest probability of classifying a node as a hotspot (S8 Table) while high centrality (eigenvector centrality) had a considerable negative effect. The small sampling size (22 nodes among which only four are hotspots, all with a small number of animals sampled, S6 Table) had a significant effect on identifying hotspots (Wilcoxon test p-value <0.05), suggesting that hotspots were more likely to be identified in nodes containing more sequences, but its effect, in most of the cases, was smaller (in magnitude) when compared to abovementioned measures (S8 Table). However, these conclusions were not statistically significant (Kruskal -Wallis test, p-values > 0.05). Due to the reduced sample size, the results of this last analysis should be regarded as very preliminary.

## Discussion

Combining pathogen genetic data with epidemiological or mobility network data can provide important insights into infectious disease dynamics [31–33]. Here we showed that genetic data

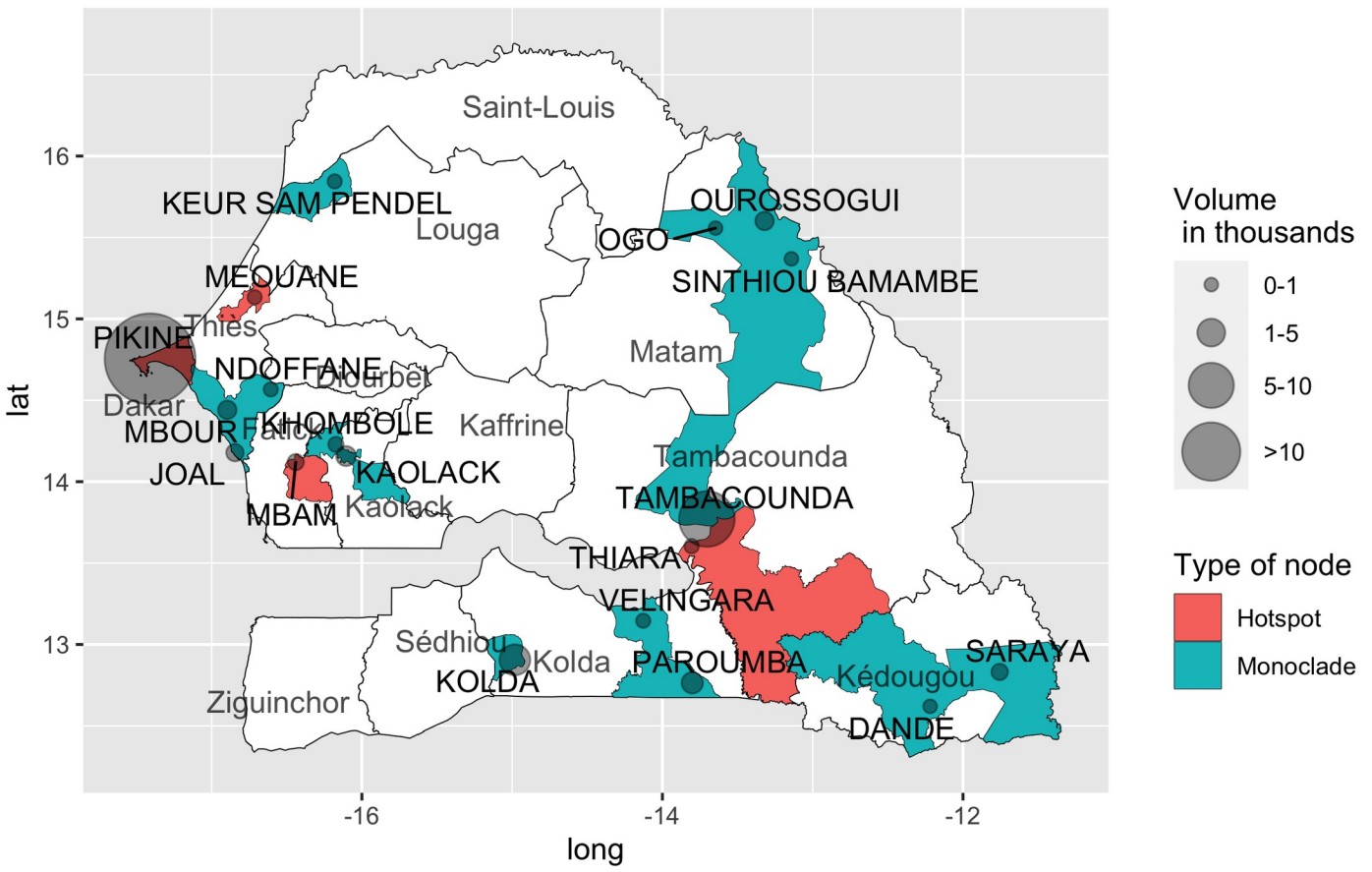

**Fig 7. Geographical locations of hotspots and monoclade.** Hotspots and monoclade are geolocalised and the wards (Administrative Unit of level 3) to which they belong are colored according to the type of node (hotspot or monoclade). Nodes are scaled based on their incoming volume of animals. Labels in uppercase letters indicate node names. Labels in lowercase letters correspond to the names of regions in Senegal. The base layer for the map used in the figure was obtained at http://www.gadm.org.

on the peste des petits ruminants virus (PPRV) combined with network analysis of animal mobility data improved our understanding of PPRV transmission dynamics in complex endemic settings. The results of the phylogenetic analyses provide evidence for this complexity in Senegal, and West Africa in general. We obtained 50 different complete PPRV N and H gene sequences, from field investigations of suspected cases of PPR in 2010–2014, mostly in Senegal. PPRV lineage II appears to be dominant in the region, with only limited presence of lineage I strains, as previously reported [6,34]. The length of sequences generated (combined length of N and H genes = 3402bp) provided enough statistical power to detect distinct clades

within lineage II circulating concurrently in the region. Some of these clades included samples from different countries, providing further support for extensive transboundary PPR circulation in West Africa [6]. Phylogenetic relationships between clades were unclear. It is possible that they would have been better resolved with full genome data, but 47/50 of the sequences obtained in this study differed in their nucleotide composition, suggesting that our level of resolution was already sufficient. Our results suggest that each clade can be considered as part of different transmission networks evolving in parallel, and this hypothesis was used as the basis to explore correlations with mobility network data.

The different correlation analyses performed showed that there was a strong correlation between virus genetic distance and both geographical and network-related distances. However, results of Mantel correlograms and Multiple regression analyses suggest that the correlation between genetic and geographical distance is strong among sites separated by relatively short distances (<100-150km), but that network connectivity best explained genetic distance patterns for larger distances. Such pattern supports the importance of animal mobility in PPRV transmission, as connections between markets separated by long distances depend on fast commercial transport that can rapidly move PPRV across the region. For shorter distances, mainly local movement of small ruminants would result in slower diffusion the virus.

When the components of the network were grouped in the shape of communities, these were presenting some pattern in rural areas and spatially fragmented in the more densely inhabited area whatever the algorithm and weighting criteria used, with a slight improvement when we used the Edge_betweenness algorithm weighted with the frequency of movements between nodes. This fragmentation of communities may be explained by the extent of long-distance animal trade and transhumance within Senegal and across countries. Notably, those patterns could depend on the production chain structure, with traders in different areas having their network of markets where they collect and sell animals before being sent to the largest consumption markets in Dakar, Kaolack or abroad [35]. Existence of such long-distance movements was further supported by our PPRV genetic data, notably the presence of identical PPRV sequences in samples from the Dakar and Matam regions (> 500 km apart), or in Tambacounda and Kolda regions (> 250 km apart), collected at an interval of only a few days (Figs 1 and 2 and S1 Table). Furthermore, the correlation between the distribution of communities and clades was strong in most scenarios when tested with Fisher's exact tests. Our results showed that viruses sampled within the same community of the network were significantly more likely to belong to the same genetic clade, providing further support for a close link between virus transmission and animal mobility.

Some nodes (villages) in the network were associated with PPRV sequences belonging to different clades (Fig 7). Such sites were tentatively considered as potential "hotspots", where multiple animal movement routes meet, increasing the risk of virus circulation and hence the infection of flocks by multiple PPRV strains of different origin at the same time. Our mobility dataset contained information on both commercial and transhumant movements without any distinction. The introduction of PPRV in an area could be related either to the purchase of animals at the market, or to the interaction between local and transhumant herders at the watering points and in grazing areas. Surprisingly, the presence of a market within the community or the administrative area did not increase the chance of a node being a hotspot. However, linear discriminant analyses showed that hotspots tended to appear at nodes with certain characteristics associated with their connectivity (frequency, betweenness, homophily). However, these trends were not significant, possibly because our dataset only included four such hotspots and our sampling effort only provides a biased snapshot of the diversity of PPRV strains circulating in the region. Further sampling to increase PPRV genetic data and possibly more targeted to network nodes with high connectivity characteristics, are needed to assess whether

such data can really help identifying nodes with a central role for PPR circulation. Another limitation in our capacity to test this hotspot hypothesis is related to the way strains were linked to mobility network nodes. Mobility data used in our analysis were mainly for intra-departmental movements, for whom health certificates are needed. Few or no information were available for small herd and local movements. To better associate strain and mobility network node, more information about livestock owner mobility and market choices should be collected through ad-hoc activities. In our case, we only considered geographical distance as association rule. However, in the analysis of hotspots, small villages appeared as hotspots. Some of these villages are located close to more renowned marketplaces (like Thiara close to Tambacounda) but they could also be resting places on the transhumance routes. Future sampling campaigns should consider collecting information on the most recent animal transaction in the farms and nearby villages, to get a clearer idea of the animals' provenance, but also of possible interactions with transhumant herds.

Based on combined virus genetic and mobility network data, this study presents the first evidence for a role for animal mobility in PPR transmission dynamics in West Africa. Correlations between PPRV genetic and mobility network data were multiple and strong. Up to now, PPR virus sampling in the field has mostly been opportunistic, based on reports of suspected disease outbreaks, leaving big gaps in our knowledge of the distribution and diversity of PPRV across the globe [6,36]. In West Africa, due to the lack of resources, there is no active surveillance system that could enable timely information about diffusion of the virus to be collected and provide a detailed description of the situation. To improve this situation, resources should be optimally allocated to areas where the risk of virus circulation is higher. Collection of animal mobility data using low-cost and easy-to-implement methodologies may provide guidance for targeted sampling in sites with higher probability of PPRV circulation, and therefore improve our evaluation of PPRV diversity and distribution in a specific region. Furthermore, combining genetic and mobility network data could help identify sites or areas that are key locations for virus entrance and spread in a region or a country. Such information would dramatically enhance our capacity to develop risk-based control and surveillance strategies that are more efficient and better adapted to local context.

## Material and methods

### Ethics statement

Veterinarians from the national veterinary services in Senegal, Guinea, Mauritania and Mali conducted the field studies as part of their routine surveillance activities, in accordance with local legislation, with no specific ethical approval required. Still, all the tissues used in the study were sourced ethically. The study was conducted in animals in contact with outdoor environments with natural exposure to diseases (PPR is endemic in the region). Ocular or nasal swabs were collected on live animals by aseptic means and/or by non-invasive methods, and tissues (lung, lymph node and/or spleen) were sampled from animals that had died of the infection.

### PPR virus sampling and analysis

The samples were kept at 4˚C during transport to the national veterinary laboratories. All samples collected were sent to CIRAD, Montpellier, France. Once there, the samples were processed in a biosafety level 3 containment laboratory.

At CIRAD, the tissue samples were cut into pieces and ground in 3 ml of Minimum Essential Media (MEM) by vortexing with 0.2 μm glass beads. The swabs were placed in 1 ml MEM and vortexed. In all cases, the sample suspensions were centrifuged for 3 min at 1000 g to

collect the supernatant. Total RNA was extracted from the supernatant using the NucleoSpin RNA virus extraction Kit (Macherey-Nagel, France), according to the manufacturer's instructions.

A RT-PCR was performed using the qScript XLT One-Step RT-PCR Kit (Quantabio, VWR, France) to amplify a 351 base pair (bp) segment of the PPRV Nucleoprotein (N) gene with the NP3/NP4 primer pair modified from Couacy-Hymman *et al.* [37] (Forward NP3: 5'-GTC-TCG-GAA-ATC-GCC-TCA-CAG-ACT-3' and Reverse NP4: 5'-CCT-CCT-CCT-GGT-CCT-CCA-GAA-TCT-3') at a final concentration of 0.6 μM. PCR was set up under the following programme: 50˚C for 30 min; 95˚C for 15 min and 40 amplification cycles (10 sec at 95˚C, 30 sec at 60˚C and 30 sec at 72˚C) and a final extension step at 72˚C for 5 min. The PCR products were resolved on 1.5% agarose gel to confirm presence of bands of expected size.

The PCR products obtained were sequenced for a preliminary phylogenetic analysis to determine the genetic lineage of the PPRV strains sampled during this study. The clean-up and sequencing of all positive PCR products in both forward and reverse directions were carried out by Cogenics (France) or Genewiz (United Kingdom). Forward and reverse DNA sequences were assembled using Geneious v. 8.1.6 and trimmed to remove poor quality portions of the sequences (final size = 255 bp). Corrected sequences were aligned with 27 PPRV N gene sequences, representative of the four PPRV genetic lineages and publicly available in GenBank (S1 Data), using Geneious v. 8.1.6. A phylogenetic tree was constructed using the maximum likelihood method as implemented in MEGA 6 [38], with node supports evaluated by bootstrap analyses (1 000 replicates).

The complete nucleoprotein (N) and hemagglutinin (H) genes were then sequenced in a selection of samples to enable more refined phylogenetic analyses. When several samples from the same location shared an identical partial N gene sequence, only one sample was used to sequence the complete N and H genes. First, custom primers were designed to amplify overlapping fragments of both genes (S9 Table). All RT-PCR were performed using the qScript XLT One-Step RT-PCR Kit (Quantabio, VWR, France), with pairs of primers at a final concentration of 0.6 μM, and the following programme: 50˚C for 30 min; 95˚C for 15 min and 40 amplification cycles (30 sec at 95˚C, 30 sec at 57˚C and 60 sec at 72˚C) and a final extension step at 72˚C for 5 min. The PCR products were resolved on 1.5% agarose gel. Complete N and H gene sequences were assembled using Geneious v. 8.1.6. N and H gene sequences were combined for each sample and aligned to thirteen sequences representatives of the four PPRV genetic lineages publicly available in GenBank, including five sequences belonging to lineage II (S2–S4 Data). The alignment was performed using ClustalW and manually curated on BioEdit v.7.2. [39]. The most suitable nucleotide substitution model for our dataset was the Kimura 2-parameter model with a discrete gamma distribution of evolutionary rates, according to Bayesian Information Criteria scores calculated in MEGA 6 [38]. This model was used to calculate genetic distance between pairs of concatenated sequences and to infer phylogenetic relationships. Phylogeny was inferred using two different methods to improve clade confidence: (1) a maximum likelihood (ML) method as implemented in MEGA 6, with node supports evaluated by bootstrap analyses (1 000 replicates), and (2) a Bayesian MCMC inference (BI) method that was performed in MRBAYES [40], with multiple runs of 10 million generations with a 20% burn-in, sampling every 100 generations, and using the default heating parameters. Based on the results of the phylogenetic analyses, sequences were clustered by "clade" according to the most ancestral tree nodes linking multiple sequences with bootstrap support >50% (ML) and posterior probability >50% (BI). Identification of these clusters was further confirmed using ClusterPicker v.1.2. [41], with node support set at 80% [42] and genetic distance at 0.07% [43]. All sequences obtained during this study have been submitted to GenBank (S2 Table).

## Animal mobility data collection and network analyses

Senegal and Mauritania have a system of certificates for livestock mobility. By law, every time a herd moves, the herders are obliged to declare the following information to the closest veterinary office: origin and destination of the movement at village level, the composition of the herd (species and number of heads), and means of transport (on foot or by vehicle). A certificate is issued to the herder, and a copy is kept by Veterinarian Officer. The copies are regularly centralised to extract information about movements in the countries concerned. For Mauritania, a national mobility survey was implemented in 2014 [11]. The survey covered 13 over 14 *Wilayas* (administrative units). In Senegal, data were collected in 2013, 2014 and 2015. However, only data from 2014 was considered in this study, as it was the most reliable available in terms of geographical and temporal coverage [14].

For both countries, we used data collected in 2014 [11,14] and only data concerning the movements of small ruminants. The villages were geo-referenced. When geo-referencing of a village was not possible, the coordinates of the centroid of the smallest administrative unit were used. The original dataset contains information on the mode of transport used, as well as the date of the movement. We removed the distinction between modes of transport and averaged and summed information over the year to estimate the average and the cumulative number of animals exchanged. We used a complex network approach, where villages corresponded to nodes and movements between nodes were represented by links weighted by the number of animals exchanged. For all the nodes we estimated some centrality measures. More specifically, we estimated for each node the number of incoming links (indegree $k_{in}$) and outgoing connections (outdegree $k_{out}$) and the distributions of these measures all over the network. For both distributions, we performed a log-log fit to check whether the distribution was following power law.

We used different types of weights to characterize a link: (i) *Frequency*—the number of times, over a period of a year, a movement took place independently of the quantity transported and means of transport; (ii) *Volume*—the total number of heads passing through the link in a year; and (iii) the *Brockmann Distance*—a measure using the annual average volume and introduced by Brockmann and Helbing [26]

$$d_{ij} = 1 - log(P_{ij})$$

where $P_{ij}$ is the fraction of outgoing animals from location *i* to *j*.

Nine network indicators were estimated to characterise the network nodes: degree, indegree, outdegree, incoming volume, outgoing volume, frequency as a destination (InFrequency), closeness, betweenness and eigenvalue centrality. Pairwise correlation analyses confirmed that these variables were uncorrelated. Community detection was done using two community detection methods: the Infomap algorithm [23], maximising information, and the Edge_Betweenness algorithm [24] to find links between communities. For each algorithm, we considered the network either unweighted (basic) or weighted using the different types of weights defined above, for a total of eight community partitions. We used a Voronoi tessellation, with centres corresponding to locations in the mobility network and with colours according to the communities detected, to identify possible spatial patterns. Two types of analysis were used to test if communities were spatially organised. The Clark Evans test [30] was used to test if communities were clustered spatially, while Join Count analysis [44] was used to check if the Voronoi cells of members of the same communities shared borders. In most cases, network nodes and virus sampling sites did not coincide. We therefore associated each PPRV sequence with the geographically closest network node, based on the assumption that most herders usually go to the closest market for their exchanges [45]. We calculated that the median distance between any virus strain sampling site and village appointed using

Voronoi diagrams was < 50 km, corresponding to 1–2 days walking distance. For samples collected in Guinea and Mali, where few or no data on mobility were available, we associated a fictitious centroid for the country as coordinates for the network nodes. Nodes found to include PPRV sequences of different clades were identified as potential "hotspots" for PPRV circulation, while nodes with only sequences from one clade were identified as "monoclade" (S6 Table).

## Correlation analyses of genetic and animal mobility data

Resistance distance, conductance variables, and Brockmann distance were estimated using CircuitScape [46]. An accessibility raster produced with the accessibility data available from http://forobs.jrc.ec.europa.eu/products/gam/ and representing the travel time from the point to a city with more than 50 000 inhabitants was used to calculate the resistance distance. Conductance calculations were based on a network circuit approach using the frequency of movements and the total number of heads as variables. The Brockman distance was considered as a resistance variable, with higher distance values corresponding to lower the probability of a passage.

We used Mantel test to determine if there was any correlation between genetic distances and the different geographical and network distance measures (S5 Data). Partial Mantel tests [27] were performed to assess whether the correlations identified were affected by interactions between network-related and geographical distances. Mantel correlograms [28] were used to detect spatial structure in the correlation patterns. The Mantel correlogram analysis is a *lagged* version of the Mantel test, where geographical distances are divided in classes (lags) defined in a way to favour homogenous repartition of samples among classes, and then correlation is evaluated within each class. The number of classes was determined using the Sturge's law, producing 7 small classes (each containing less than 8 pairs) and 4 large ones (containing between 10 and 31 pairs). Half of the pairs were contained in the first 2 classes. For all correlation analyses, significance was evaluate using permutation tests (1000 permutations). Multiple Regression on distance matrices (MRM) analysis was performed to assess the relative importance of each explanatory distances (geographical *vs* network-related) [28,29]. MRM is a generalisation of the multiple regression approach and partial Mantel test: distance matrices are permuted in a similar way as for the mantel test, and regression coefficients for each explaining matrix are estimated at each permutation [47].

We looked for factors that could explain the spatial distribution of clades. Networks are characterised by the presence of structures, like communities. These correspond to tightly connected set of nodes whose elements interact more within (i.e. with other members of the same community) that with other nodes of the network [22]. The presence of communities in the network could indicate the existence of underlying dynamics in network formation that favour the creation of links among set of nodes. In this work we used two algorithms to detect communities: (i) The information-based InfoMap algorithm that finds modules in the network exploiting its regularities [48] and reduces the amount of information needed characterize it, and (ii) the edge-betweenness algorithm that removes links with higher betweenness and estimates modularity iteratively based on the assumptions that links with high betweenness are those connecting communities [49].

The modularity of the network, i.e. the number of links existing among members of the same community in comparison with those with nodes of other communities, is defined as

$$Q = \frac{1}{2m} \sum_{ij} \left( A_{ij} - \frac{k_i k_j}{2m} \right) \delta\left( c_i, c_j \right)$$

where the sum is extended over all nodes (*ij*) [50]. $A_{ij}$ is the adjacency matrix element between the nodes whose values are 0 or 1, $\frac{k_i k_j}{2m}$ is the expected number of edges between nodes if links

were disposed randomly, and $\delta(c_i, c_j)$ is 1 if two nodes belong to same community, 0 otherwise. The higher is the modularity, the better is the partition. When the network is weighted, the intensity of the connections among members of the same community can also be considered. Here, the two algorithms have been used considering both for the unweighted and weighted cases, resulting in several partitions. To assess the goodness of each partition and the similarity between community partition, we use the modularity and the Rand Index respectively [51]. For each possible pair of nodes in the network, the Rand index evaluates how many pairs belong to the same community in the two partitions. The higher the value of the Rand Index, the higher the similarity between partitions. The community detection algorithms detected several communities, of which only a few contained nodes where the virus sampled, here termed "PPRV-community". Our analysis focused only on the set of these communities. We checked for a correlation between being in the same PPRV community and in the same clade. For each community partition, we used a Fisher's exact test to check for a relation between community partition and clade distribution, i.e. if the presence of communities could factor the dominance on a clade in a set of nodes. We tested the hypothesis that two sequences were more likely to be in the same clade if they were in the same community by introducing some binary variables whose prefix *same* indicates if two nodes include PPRV sequences of the same clade (1) or not (0) and if they are (1) or not (0) in the same community. The odds ratio was computed for the two groups and the Fisher Exact test used to check whether the OR was different from 1 and to estimate its p-value and 95% Confidence Interval.

## Characterisation of hotspots

We used the terms hotspot to indicate nodes where PPRV sequences of different clades were found, while we used the term monoclade to indicate nodes where all the sequences available belonged to the same clade (S6 Table). We used linear discriminant analysis, using the nodes' network characteristics, to identify the factors that characterise hotspots. In a preliminary analysis we found that more sequences were collected in hotspot nodes compared to monoclade ones, and the limitedness of the sampling procedure could result in misclassification of nodes. To take this into account, we considered the number of sequences available as a possible factor that could characterise hotspot. Then, we included other possible factors, related to livestock trades and mobility, which could help identifying possible hotspot locations. First, we assessed if the presence of a market explained the presence of "hotspots" of virus genetic diversity. Data on the location of markets were kindly provided by the Senegalese Veterinary Services. We first checked at administrative Units of Commune (Administrative level 3) and Department (administrative level 2) if markets were present, and second, if markets were present within a radius of 50 and 100 km from the virus sampling location. We used a relative risk approach considering the presence or not of markets (at a certain level) as risk factor. The analysis was restricted only to those nodes in whose vicinity strains were collected. In this context, we also tested if homophily, i.e. the tendency of a node to create links with nodes in the same community, was a relevant factor characterising hotspots. For each of those nodes we estimated the "bare" homophily, estimating only the number of connections shared with other nodes in the same communities, but also the strength of the homophily, weighting the connections by the frequency and the volume exchanged.

All analyses were performed using R v.3.4.0.

## Supporting information

**S1 Fig. PPR partial N gene phylogenetic analysis.** Phylogenetic tree constructed using a maximum likelihood inference method and showing the relationship based on partial N gene

sequences (255 bp) of peste des petits ruminants virus (PPRV) samples obtained in this study and publicly available sequences representative of all 4 PPRV genetic lineages. Samples collected in this study are indicated by symbols according to sampling location (Guinea, *graphic object* Mali, Mauritania, *graphic object* Senegal). The numbers at the nodes are bootstrap values obtained from 1 000 replicates (Maximum Likelihood methods). Bootstrap values are shown if $> 50\%$.
(TIF)

**S2 Fig. Phylogenetic tree described in Fig 1 showing putative transmission clusters identified using ClusterPicker with node support and genetic distance thresholds at 80% and 0.07%, respectively.**
(TIFF)

**S3 Fig.** Indegree (A), outdegree (B) and frequency of the link (C) distribution in log-log scale and the link's weight cumulative distribution in log scale (D). Shaded areas correspond to the results of linear regression, the dashed lines correspond to the 50th and 90th percentiles.
(TIFF)

**S4 Fig.** Mantel correlogram for all spatial distances considered (column), with (triangles) and without (circle) control by network distances (line). Color corresponds to significant and non-significant Mantel coefficients.
(TIFF)

**S5 Fig.** Community size distribution, based on the community detection algorithm used (row) and the link's weight definition (column).
(TIFF)

**S6 Fig.** Community members' distance distribution based on the community detection algorithm used (row) and the link's weight definition (column). Colours corresponds to the community index. Each boxplot shows the distribution of geographical distance among locations belonging to the same community.
(TIFF)

**S7 Fig.** Correlation between community size and number of clades based on the community detection algorithm used (row) and the link's weight definition (column).
(TIFF)

**S1 Table. List of samples collected and tested for Peste des Petits Ruminants infection.**
(DOCX)

**S2 Table. List of PPRV nucleoprotein and hemagglutinin gene sequences obtained in this study.**
(DOCX)

**S3 Table. Correlation coefficients between genetic distance and spatial and network measures.**
(DOCX)

**S4 Table. Partial Mantel tests of correlation coefficients between genetic distance and spatial and network measures.**
(DOCX)

**S5 Table. Results of the Multiple Regression on distance matrices analyses (MRM).**
(DOCX)

**S6 Table. Number of sequences available around a network node and type of node.**
(DOCX)

**S7 Table. Average values of network characteristics in hotspot and monoclade nodes.**
(DOCX)

**S8 Table. Results of Linear Discriminant Analysis.**
(DOCX)

**S9 Table. Description of primers used to sequence the complete PPRV nucleoprotein and hemagglutinin genes.**
(DOCX)

**S1 Data. PPRV partial N gene alignment in fasta format.**
(TXT)

**S2 Data. PPRV complete N gene alignment in fasta format.**
(TXT)

**S3 Data. PPRV complete H gene alignment in fasta format.**
(TXT)

**S4 Data. PPRV complete concatenated N and H genes alignment used in phylogenetic analyses.**
(TXT)

**S5 Data. Dataset of pairwise distances including genetic, geographical, network-related distances in cvs format.**
(CSV)

## Acknowledgments

The authors would like to thank the staff of the UMR ASTRE (CIRAD), the Laboratoire Central Vétérinaire (Mali), the Laboratoire National d'Elevage et de Recherches Vétérinaires (Senegal), Direction des Services Vétérinaires (Senegal), the Office National de Recherches et de Développement de l'Elevage (Mauritania), and the Institut Supérieur des Sciences et de Médecine Vétérinaire (Guinea) for their help with this project. Francesca Fagandini Ruiz (CIRAD) helped in obtaining road distances used in the correlation analyses. A.A. would thank Stephan Hue (LSHTM) and Roberto Molinari (Auburn University) for fruitful discussions.

## Author Contributions

**Conceptualization:** Arnaud Bataille, Geneviève Libeau, Andrea Apolloni.

**Data curation:** Arnaud Bataille, Ismaila Seck, Baba Sall, Coumba Faye, Mbargou Lo, Ahmed Bezeid El Mamy, Caroline Coste, Cécile Squarzoni Diaw, Olivier Kwiatek, Andrea Apolloni.

**Formal analysis:** Arnaud Bataille, Habib Salami, Caroline Coste, Olivier Kwiatek, Andrea Apolloni.

**Funding acquisition:** Arnaud Bataille, Modou Moustapha Lo, Geneviève Libeau, Andrea Apolloni.

**Investigation:** Habib Salami, Ismaila Seck, Modou Moustapha Lo, Aminata Ba, Mariame Diop, Baba Sall, Coumba Faye, Mbargou Lo, Lanceï Kaba, Youssouf Sidime, Mohamed

Keyra, Alpha Oumar Sily Diallo, Mamadou Niang, Cheick Abou Kounta Sidibe, Amadou Sery, Martin Dakouo, Ahmed Bezeid El Mamy, Ahmed Salem El Arbi, Yahya Barry, Ekaterina Isselmou, Habiboullah Habiboullah, Abdellahi Salem Lella, Baba Doumbia, Mohamed Baba Gueya, Caroline Coste, Cécile Squarzoni Diaw, Olivier Kwiatek, Andrea Apolloni.

**Methodology:** Arnaud Bataille, Habib Salami, Cheick Abou Kounta Sidibe, Caroline Coste, Cécile Squarzoni Diaw, Olivier Kwiatek, Andrea Apolloni.

**Supervision:** Arnaud Bataille, Lanceï Kaba, Mamadou Niang, Ahmed Bezeid El Mamy, Geneviève Libeau, Andrea Apolloni.

**Validation:** Arnaud Bataille, Cécile Squarzoni Diaw, Olivier Kwiatek, Andrea Apolloni.

**Visualization:** Arnaud Bataille, Caroline Coste, Andrea Apolloni.

**Writing – original draft:** Arnaud Bataille, Andrea Apolloni.

**Writing – review & editing:** Habib Salami, Ismaila Seck, Modou Moustapha Lo, Mamadou Niang, Cécile Squarzoni Diaw, Geneviève Libeau.

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
