## [Decision Letter · Decision Letter 0]

11 Aug 2020

Dear Mr Bataille,

Thank you very much for submitting your manuscript "Combining viral genetic and animal mobility network data to unravel peste des petits ruminants transmission dynamics in West Africa" for consideration at PLOS Pathogens. As with all papers reviewed by the journal, your manuscript was reviewed by members of the editorial board and by several independent reviewers. In light of the reviews (below this email), we would like to invite the resubmission of a significantly-revised version that takes into account all of the reviewers' comments. Please consider this a major revision requiring more than just editorial changes or discussion points.

While the reviewers have recognised the importance and relevance of your study, they raised major concerns about the soundness of the phylogenetic analyses and the appropriateness of the statistical tests you used. Please address all their comments. We cannot make any decision about publication until we have seen the revised manuscript and your response to the reviewers' comments. Your revised manuscript will be sent back to reviewers for further evaluation.

Sincerely,

Guillaume Fournié

Guest Editor

PLOS Pathogens

Marco Vignuzzi

Section Editor

PLOS Pathogens

Kasturi Haldar

Editor-in-Chief

PLOS Pathogens

orcid.org/0000-0001-5065-158X

Michael Malim

Editor-in-Chief

PLOS Pathogens

orcid.org/0000-0002-7699-2064

Reviewer's Responses to Questions

**Part I - Summary**

Reviewer #1: In this study, Baitaille and colleagues using newly sequenced PPRV genetic data and animal movement data investigate the patterns of PPRV circulation, mostly in Senegal, between 2010-2014. Key findings include finding co-circulation of multiple viral lineages in Senegal and strong correlation between viral genetic distances and network-related distances. In addition, they also identified potential “hotspots” of PPRV transmission, which were characterised by the detection of multiple viral lineages. Network analysis indicated that these hotspots correspond to greater connectivity and movement.

The motivation of this study is clearly communicated in the introduction, and notably the authors generated 50 complete nucleocapsid and haemagglutinin gene sequences to investigate PPRV transmission in Senegal and neighbouring countries. While the sequencing effort is admirable, especially we have limited information especially as PPRV diversity appears comparatively undersampled despite being an important threat to food security and economy, and current sequences on Genbank appear to be relatively short (~300 bp in length), I do have concerns about their phylogenetic analyses. In particular, given the limited number of sequences and their low genetic diversity, I am not sure if the authors have detected six distinct clades or just one single clade/lineage. As a consequence, this may affect some of their key findings (e.g., locations where multiple lineages detected).

Reviewer #2: Authors used N and H genes concatenated PPRV Phylogenetic relationships between clades which are unclear and authors accepted this facts in their manuscript. Analysis of full genome of PPRV may would have been better resolved. Therefore authors' hypothesis suggesting that each clade can be considered as part of different transmission networks is not convincing. A clear conclusion of work is not possible for the readers. Further PPRV exists in one serotype and 4 genetic lineages. Authors found mainly lineage II PPRV circulation in West Africa although some Lineage I circulation was found. Evolution rate is very very low in PPRV as reported in many recent studies and main changes are seen in P gene. Therefore it is not convincing that each clade can be considered as part of different transmission networks. Analysis of full genome may be helpful, however as said above due to low evolutionary rate this may also be difficult.

Reviewer #3: Thank you for inviting me to review this paper. This research represents a unique approach to understand the drivers of PPRV circulation in a low-resource endemic setting. The data utilized is unique in that it combines genetic data with animal mobility networks, the latter of which is seldom available in low resource settings. The approach is relatively sound, and makes an important contribution to the field. However, I do have a number of major and minor comments. For the major comments, I have a few concerns about the choice of statistical tests. In addition, some of the major conclusions are not well-supported by the data presented. For example, the analysis presented is not convincing that network-based connections are more important correlates of PPRV genetic distance than geographic distance. The correlations look nearly identical. Second, villages are labeled as hotspots if >1 clade was identified. However, the data set and sampling effort is not extensive (and likely not homogenous across locations), and no statistical support is presented for having >1 versus only 1 clade. Given that a number of subsequent analyses are based on this classification, it is important to cement this assertion.

**Part II – Major Issues: Key Experiments Required for Acceptance**

Reviewer #1: 1) The authors identified six clades from their 50 PPRV sequences. However, on inspection of the phylogeny in Figure 1, these clades appear to genetically very similar. Therefore, it is unclear if these clades truly represent distinct lineages or they are just one single lineage (lineage II), which have been widely distributed. As this result/observation is crucial for some of the downstream analyses, I think the authors need to provide better justification/support for the six identified clades (e.g., by genetic distance or time to most recent common ancestor). There are methods for identifying transmission clusters using phylogenetics, particularly in HIV-1 and HCV epidemiology. Therefore, I think it would be worthwhile looking at these approaches (e.g., https://academic.oup.com/ve/article/5/2/vez039/5584304, https://bmcbioinformatics.biomedcentral.com/articles/10.1186/1471-2105-14-317) when identifying transmission clusters of PPRV.

2) The results of the phylogenetic analyses need to be better explained in the main text. How many sequences were found per clade? What was their phylogenetic support? How was the main alignment curated? Did you use Genbank to include complete N and H PPRV sequences from relevant countries? How many sequences were included in the final alignment and how many new sequences did your study contribute? I was left wanting more detail about the analysis.

3) Given it is a central result, I would like to see the correlation of genetic distance against spatial and network-related distances as a main figure, in addition to the statistical test. As there are only a limited number of sequences analysed in this study, I found it difficult to ascertain from the current analysis how many data points contribution in this correlation analysis. Likewise, I would also like to see the correlation plots for community size and the number of clades observed in these communities.

4) As most readers of PLoS Pathogens are unlikely to be familiar with the network analyses presented in this study, it would be helpful to have a short description/summary of the main methods used to infer networks/communities in the main text. It might be worthwhile to include a figure to illustrate what the authors mean by communities, and how related this information to virus genetic data to identify hotspots of transmission.

5) The results presented in Figure 3 is difficult to interpret, especially as someone with limited expertise in network analyses. The summary figure as suggested in the previous point might help address some of these concerns. However, I am still unclear what the colour labels for community in this figure correspond to. I appreciate panels A and B are showing results from the two different methods, but it is unclear if these results are showing a similar pattern or not.

6) The cyan shading in Figure 3 is very jarring, especially when you have similarly coloured circles overlayed on top. Is this colour scheme necessary?

7) Why was the mobility data from only 2014 selected and analysed. If this is due to availability of data, please clarify. If not, does the main findings hold when mobility data from other years are analysed instead?

Reviewer #3: L218-219: The authors write that genetic distance was more correlated to network distances ( than geographic distances. However, Table S2 shows that all distance with the exception of volume conductance ( a network distance), has similar correlations (between 0.72-0.79). So this statement is not supported by the data presented. This carries over to the discussion (L303-305)

L224. Its unclear to me that MR-QAP is the appropriate approach for these data. MR-QAP is based on ordinary least squares, so the distribution of the outcome (genetic distance) should be someone normal. It would be beneficial for the authors to present this information. If the netlm assumptions are violated, it might be possible to dichomotmize the outcome and use netlogit. Also, which network and geographic distances were included in the regression. All of them? Are these variables co-linear, and how might that impact the outcome? The coefficient values are extremely small, which suggests that the x-variables were large. It might be beneficial to standardize these variables, or potentially put them on a log scale. Alternatively, consider created three levels.

L230. Here, the authors state that communities cluster spatially, but later in the paragraph they state there is no clear spatial pattern. Related to this, Figures 3 and 4 show that there is a great deal of variability in the community assignments, which begs the question if the community structure is somewhat weak and therefore inconsistent. It may help to report modularity scores.

L248. The word “risk” comes across oddly here. My understanding is that the authors aimed to determine if any two sequences were more likely to be in the same clade if they were part of the same community. I would not label this as “risk,” rather that the distribution of clades across communities was non-homogenous. Why not use odds/odds ratio instead of risk? It also would make sense to construct a 2x2 contingency table here, with same community (0/1) and same clade (0/1), and perform the Fisher’s exact test. However, they also performed a t-test, and it is unclear what that was testing. Clarification is needed.

L256-258. I am not entirely convinced that these “hotspots” are true hotspots based on simply having >1 clade identified. This should be backed up with statistics. The paper reports only 39 sequences across the network of hundreds of nodes. Thus, are nodes sufficiently sampled to determine the absence of other clades at poorly sampled nodes?

L272-274. This is an interesting hypothesis. However, I would expect that having more between-community ties (not within-community) would be more important for introducing diverse clades (i.e., the “strength of weak ties” in shaping disease spread, where weak ties are between-communities). Here, the data shows that hotspot nodes have a tendency to form more within-community ties. However, is it important that they are within-cluster (homophily), or do the high homophily values simply reflect the higher overall degree of hotspot nodes. I suggest reporting both the strength of within-community and between-community ties to help tease apart degree, and within/between community ties.

**Part III – Minor Issues: Editorial and Data Presentation Modifications**

Reviewer #1: 1) Looking at the fasta files and Table S2, there appears to be 50 sequences that were generated in this study (not 49 as indicated in the main text), which includes 36 sequences from Senegal (not 39 as indicated in the main text).

2) Please provide a brief description of quadratic assignment procedure.

3) Figure 3 – axes are labelled simply ‘x’ and ‘y’. To be honest, the long/lat axes could be removed in this Figure as well as from Figure 4.

Reviewer #3: L45: Revise to “significantly more likely”

L127: Change “completed” to “complemented”

L184: The phrase “at least 9 800 animals annually” falls awkwardly in the sentence. Revise.

L185. The authors write that many links occurred “once a year.” My understanding is that data was only collected for a single year, and “once a year” sounds as if the links are repetitive across years, which is not supported by the study design.

L194-195. This sentence is repetitive with the previous. I suggest deleting.

L200. The authors first write that community size and structure differed based on weighting and algorithm. However, here they state the similarlity between community structure was high. Here, its not clear what is being compared.

L209. Revise to “others”

L210. Define brockman distance.

L216. What is included in “other distance.” Spatial distances, or network-related distance?

L219. The Mantel correlation between genetic distances and what?

L232. Revise to “the largest community detected by the Edge_Betweenness algorithm, was not spatially clustered regardless of weighting”

L243. The authors should use Spearman correlation for this.

L269. There is no Table S9. Additionally, the text implies that frequency is an incoming metric, but that is unclear in Table S4. In addition, hotspot nodes have higher indegree, but not incoming frequency?

L269. Comparison of node metrics should be backed up with a statistical test (Kruskal-Wallis would be appropriate).

Tables S4-S5. Check numbering. The text and the numbering may not be consistent? Or perhaps the text in the results needs greater clarification to understand how the results relate to the tables. For example, L275 says that Table S5 is a discriminant analysis… however it appears from the table captions that S5 is hierarchical clustering and S4 is a linear discriminant analysis.

L294-294. The authors report multiple co-circulating clades. While I agree with their previous description that these likely represent common transmission chains and a recent common ancestor, it would be helpful to know what the genetic distance was for these clades.

L296. “Phylogenetic relationships between clades were unclear.” Is this because of low bootstrap support?

L428-429. Its unclear why clades were identified using both ML and a Bayesian method. I would have expected that final inference would be done with one or the other (with ML often a precurser to Bayesian analysis). The authors should elaborate more on this.

L448. Remove the word “complex.” The SNA approach appears to be a standard weighted network.

L472-473. It appears that viral sequences were assigned, in some cases, to the closest node for which there was data in the network. The median distance corresponded to 1-2 days of walking. How sure are the authors that animals that far away may not go to a different node? At the very least, this assumption should be mentioned in the discussion.

L505-506. “The assumption to test was if communities are monoclade or a hotspot.” Revise for clarity.

PLOS authors have the option to publish the peer review history of their article (what does this mean?). If published, this will include your full peer review and any attached files.

Reviewer #1: No

Reviewer #2: No

Reviewer #3: No
---

## [Decision Letter · Decision Letter 1]

10 Jan 2021

Dear Mr Bataille,

Thank you very much for submitting your manuscript "Combining viral genetic and animal mobility network data to unravel peste des petits ruminants transmission dynamics in West Africa" for consideration at PLOS Pathogens. As with all papers reviewed by the journal, your manuscript was reviewed by members of the editorial board and by independent reviewers. Based on these reviews, we are likely to accept this manuscript for publication, providing that you modify the manuscript according to the review recommendations (see below).

Sincerely,

Guillaume Fournié

Guest Editor

PLOS Pathogens

Marco Vignuzzi

Section Editor

PLOS Pathogens

Kasturi Haldar

Editor-in-Chief

PLOS Pathogens

orcid.org/0000-0001-5065-158X

Michael Malim

Editor-in-Chief

PLOS Pathogens

orcid.org/0000-0002-7699-2064

Comments needing to be addressed:

Reviewers’ comments have been thoroughly addressed. There are a few more minor comments below:

On Line 199-201, you write “Both the indegree and outdegree distributions followed a power law indicating the presence of a few highly connected nodes (hubs) and many poorly connected nodes (Figure S3).” You do not report the power law coefficient and you do not seem to have mentioned the fitting procedure in the methods. Please add this information. Also, you have not tested statistically for power law behaviour of your degree distributions. The plot might also be interpreted as this behaviour not being necessarily obvious. Please do so or nuance your statement.

On line 234-6, you wrote “Only genetic distances between pairs of sequences within well-defined phylogenetic clades were used for these analyses, as they represented groups of viruses originating from a common chain of transmission events.” It would be useful to mention how many pairs are considered. Likewise, for the Mantel correlogram analysis, how many pairs are included under each class?

It is unclear how the hotspot analysis is influenced by the number of sequences available in each node. It would be helpful for the readers to summarise the distribution of sequences/site, especially the number of sites where multiple sequences are available, and whether being a hotspot vs monoclade site is associated with the number of sequences available per site.

On line 612, you wrote “odds ratio was then computed for the two groups using Fisher Exact test.” This is confusing as you do not need to conduct Fisher exact tests in order to compute Odds Ratio. To avoid confusion, please specify explicitly the actual equations of the odds ratios you computed.

On Table S6, hotpost and monoclade are defined based on the number of clades to which the sequences sampled in those sites belong. The purpose of the hierarchical clustering analysis mentioned in the title of Table S6 is therefore unclear.

On line257, specify the actual range of distances for both subsets.

Typos:

Table S4 caption: “distance distances” remove “distance”

On line 215, you refer to Figure S4, should it be Figure S5?

On line 267, “closely related PPRV strains in the same clade” replace “in” by “of”

The Figure numbers are missing in several places (Line 275, 281, 283, 298). Please check your manuscript carefully

Reviewer Comments (if any, and for reference):

Reviewer's Responses to Questions

**Part I - Summary**

Reviewer #1: The authors focus on an important pathogen that poses a significant threat to food security and global health. The introduction to their primary objective; to combine viral genetic and social network data to gain a better understanding of PPRV transmission dynamics in an endemic region, is clear and convincing. By providing additional details about the phylogenetic analysis and sequence data, the authors have addressed most of my previous concerns. Although their newly generated sequence data of complete N and H gene sequences is relatively small (50 sequences), their combined gene lengths should provide resolution to identify distinct clades within a lineage. This study demonstrates that by integrating mobility and genetic data, we can obtain novel insights about virus transmission. This is an important step for PPRV transmission, but likely for other pathogens that circulate in livestock/wild animals and where information is available on their movement patterns.

Overall, the revised manuscript is well written and likely to be of interest to the readership of PLoS Pathogens.

**Part II – Major Issues: Key Experiments Required for Acceptance**

Reviewer #1: I have no major concerns.

**Part III – Minor Issues: Editorial and Data Presentation Modifications**

Reviewer #1: Line 281 and 298: There are a few instances in the text where figure number is missing - please double-check.

Line 332-334 "This is one of the most complete PPRV genetic datasets ever obtained from a single endemic region" - I appreciate that PPRV sequences are scant, and the authors have taken commendable efforts in generating the sequence data, but given that the size of the dataset is small and the likely sample bias, this statement is a bit misleading or at the very least exaggerated. I think the authors should emphasize the fact that by using complete N and H sequences, with a combined length ~3500, they had sufficient statistical power to detect distinct clades within lineage II. I believe this is a particular strength of the study, as it is unlikely that the authors would have been able to obtain the same insights if they used short gene sequences.

Figure 1 - the inset figure is not easy to read. In particular, I could not discern the black and turquoise diamonds. Based on the high-res figure, should the turquoise diamonds be grey?

Figure 7 isn't mentioned in the main text (I suspect this may be related to my first point). I also have issues with the presentation of the figure. I found the text size in this figure too small and as a result challenging to read. Furthermore, compared to the country maps presented in Figures 2, 5, and 6, the map in Figure 7 seems elongated along the latitude axis.

To simplify this figure, I would suggest colouring in each region depending on whether there are multiple or single lineages detected, instead of using small points. If you want to highlight the hotspots in more detail, you could present a zoomed-in version for these locations.

PLOS authors have the option to publish the peer review history of their article (what does this mean?). If published, this will include your full peer review and any attached files.

Reviewer #1: No
---

## [Editor Report · Decision Letter 2]

2 Feb 2021

Dear Mr Bataille,

Thank you very much for submitting your manuscript "Combining viral genetic and animal mobility network data to unravel peste des petits ruminants transmission dynamics in West Africa" for consideration at PLOS Pathogens. We will accept this manuscript for publication, however, before formal acceptance we ask that you address a few remaining points, this will not require any additional round of reviews.

The following comments need to be addressed:

Figure 7: The latitude axis is still elongated.

Hierarchical clustering in Table S7: this is still unclear, as it seems that Table S7 only provides average values of statistics for two pre-defined sets of nodes, and no hierarchical clustering analysis has actually been done. If you did a hierarchical clustering analysis, please explain it in the methods.

Table S7: in addition to average values, please provide a range or standard deviations.

There seems to be an issue with the formatting of Tables S7-8, please edit these.

Sincerely,

Guillaume Fournié

Guest Editor

PLOS Pathogens

Marco Vignuzzi

Section Editor

PLOS Pathogens

Kasturi Haldar

Editor-in-Chief

PLOS Pathogens

orcid.org/0000-0001-5065-158X

Michael Malim

Editor-in-Chief

PLOS Pathogens

orcid.org/0000-0002-7699-2064

The following comments need to be addressed:

Figure 7: The latitude axis is still elongated.

Hierarchical clustering in Table S7: this is still unclear, as it seems that Table S7 only provides average values of statistics for two pre-defined sets of nodes, and no hierarchical clustering analysis has actually been done. If you did a hierarchical clustering analysis, please explain it in the methods.

Table S7: in addition to average values, please provide a range or standard deviations.

There seems to be an issue with the format of Tables S7-8, please edit these.
---

## [Editor Report · Decision Letter 3]

17 Feb 2021

Dear Mr Bataille,

We are pleased to inform you that your manuscript 'Combining viral genetic and animal mobility network data to unravel peste des petits ruminants transmission dynamics in West Africa' has been provisionally accepted for publication in PLOS Pathogens.

Best regards,

Guillaume Fournié

Guest Editor

PLOS Pathogens

Marco Vignuzzi

Section Editor

PLOS Pathogens

Kasturi Haldar

Editor-in-Chief

PLOS Pathogens

orcid.org/0000-0001-5065-158X

Michael Malim

Editor-in-Chief

PLOS Pathogens

orcid.org/0000-0002-7699-2064

---

## [Editor Report · Acceptance letter]

10 Mar 2021

Dear Mr Bataille,

We are delighted to inform you that your manuscript, "Combining viral genetic and animal mobility network data to unravel peste des petits ruminants transmission dynamics in West Africa," has been formally accepted for publication in PLOS Pathogens.

Best regards,

Kasturi Haldar

Editor-in-Chief

PLOS Pathogens

orcid.org/0000-0001-5065-158X

Michael Malim

Editor-in-Chief

PLOS Pathogens

orcid.org/0000-0002-7699-2064